# Engineering recurrent neural networks from task-relevant manifolds and dynamics

**Eli Pollock**[ID]*, **Mehrdad Jazayeri**[ID]*

Department of Brain & Cognitive Sciences, McGovern Institute for Brain Research, Massachusetts Institute of Technology, Cambridge, Massachusetts, United States of America

* epollock@mit.edu (EP); mjaz@mit.edu (MJ)

## Abstract

Many cognitive processes involve transformations of distributed representations in neural populations, creating a need for population-level models. Recurrent neural network models fulfill this need, but there are many open questions about how their connectivity gives rise to dynamics that solve a task. Here, we present a method for finding the connectivity of networks for which the dynamics are specified to solve a task in an interpretable way. We apply our method to a working memory task by synthesizing a network that implements a drift-diffusion process over a ring-shaped manifold. We also use our method to demonstrate how inputs can be used to control network dynamics for cognitive flexibility and explore the relationship between representation geometry and network capacity. Our work fits within the broader context of understanding neural computations as dynamics over relatively low-dimensional manifolds formed by correlated patterns of neurons.

## Author summary

Neurons in the brain form intricate networks that can produce a vast array of activity patterns. To support goal-directed behavior, the brain must adjust the connections between neurons so that network dynamics can perform desirable computations on behaviorally relevant variables. A fundamental goal in computational neuroscience is to provide an understanding of how network connectivity aligns the dynamics in the brain to the dynamics needed to track those variables. Here, we develop a mathematical framework for creating recurrent neural network models that can address this problem. Specifically, we derive a set of linear equations that constrain the connectivity to afford a direct mapping of task-relevant dynamics onto network activity. We demonstrate the utility of this technique by creating and analyzing a set of network models that can perform a simple working memory task. We then extend the approach to show how additional constraints can furnish networks whose dynamics are controlled flexibly by external inputs. Finally, we exploit the flexibility of this technique to explore the robustness and capacity limitations of recurrent networks. This network synthesis method provides a powerful means for generating and validating hypotheses about how task-relevant computations can emerge from network dynamics.

**Data Availability Statement:** Code for reproducing the results of this paper can be found at Github (https://github.com/elipollock/EMPJ).

**Funding:** MJ was supported by a CRCNS grant funded through the National Institute of Health

(NIMH: 1R01MH122025-01) and the McGovern Institute. The funders had no role in study design, data collection and analysis, decision to publish, or preparation of the manuscript.

**Competing interests:** The authors have declared that no competing interests exist.

## Introduction

Early studies of the link between the brain and behavior focused on simple sensory and motor functions and documented how single neurons encode variables such as visual orientation [1] and movement endpoint [2]. More recently, the field has begun to move toward richer tasks in which the stimulus-response relationships are more complex [3] and can vary with unobservable ("latent") internal variables. For instance, in decision-making tasks, reaction times depend on a latent decision threshold that is adjusted internally based on prior expectations and costs [4].

However, it is not known how neural systems substantiate latent variables. Initial efforts to tackle this question found subsets of single neurons in various higher order brain areas whose activity was modulated by behaviorally-inferred latent variables [5]. For example, certain neurons in the lateral intraparietal cortex carry signals related to covert spatial attention, motor intention, and decision variables [6]. However, recent large-scale recordings suggest that latent variables are encoded by patterns of activity across populations of neurons [7] that emerge through interactions between neurons [8]. Therefore, it is necessary to develop a framework to explore how recurrent interactions between neurons give rise to latent variables and use those variables to perform task-relevant computations.

Recent advances in machine learning have made it possible to examine this problem using recurrent neural network (RNN) models [9,10]. For example, one common approach is to create RNN models that are optimized to perform specific tasks [11–17] or emulate specific patterns of neural activity [9,18–20], and use those models to test hypotheses in relation to neural and behavioral data. Task-optimized RNN models can also be reverse engineered using tools from dynamical systems theory to explore how they perform specific tasks [11–14,16]. However, the RNN solutions that such optimization techniques find are not constrained to match any specific latent variable model of the behavior, and therefore, there is no guarantee that they can be used to reject or validate such latent models. Here, we propose a novel analysis-by-synthesis approach that addresses the inverse problem of engineering RNNs that implement a desired latent variable model [21,22]. We develop the methodology within the context of a simple working memory task, demonstrate how a single RNN can use inputs to flexibly choose between a family of latent models, and explore how the mapping from latent models to RNNs impact network performance and connectivity.

## Results

### A state-space framework for describing latent models of behavior

The basic objective of behavioral latent models is to explain the relationship between observable task variables and a set of hypothesized latent variables. This relationship can be captured using a state-space framework [23] (Fig 1). In this framework, each variable is represented as one dimension in a coordinate system, which we refer to as latent task space. We invoke the term "latent" to note that the model includes variables that are not observable. Every point in this space is a unique combination of latent and observed task variables. The collection of points in this space covers the entire range of behaviors that a latent model could conceivably capture. In most cases of interest, only certain regions of the latent task space are needed to capture behavior in a given task. We will refer to this subregion as the latent task manifold. Finally, changes of the latent and observable variables over time can be depicted as trajectories in the latent task manifold. A successful model should additionally capture the constraints that govern these dynamics, which we refer to as "latent task dynamics."

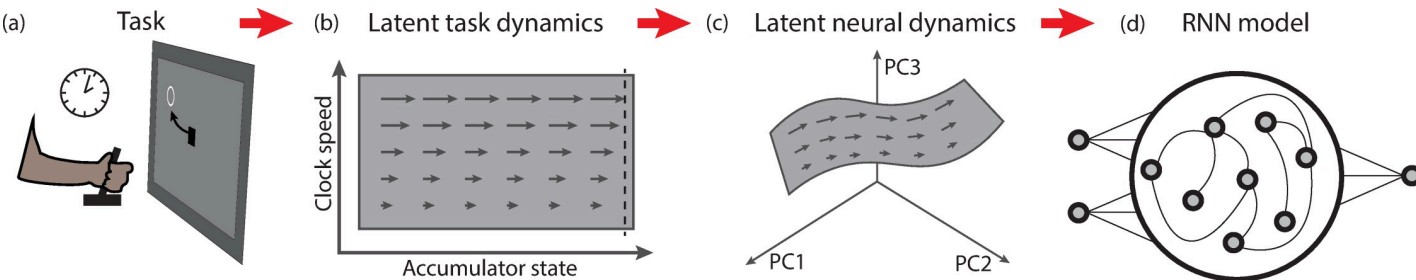

**Fig 1. Embedding latent task dynamics into recurrent neural networks.** a) A hypothetical task, adapted from [27]. The subject is instructed to produce a time interval by making a delayed movement. b) Latent task space over a manifold (gray rectangle) based on a clock-accumulator model [28]. The y-value reflects the speed of the clock, and the x-value reflects the state of the accumulator before movement initiation. In this model, the instruction sets the speed of the clock (length of the arrow) at which the accumulator evolves over time (faster for shorter intervals). Movement is initiated when the accumulator reaches a threshold (dashed line). c) A nonlinear embedding of the task manifold depicted in a neural state space spanned by the first three principal components of population activity. d) An RNN model to establish the desired nonlinear embedding.

## Mapping latent task space to latent neural space

An analogous state-space framework can be used to capture the organization of neural activity within RNN models. The neural state space for a given RNN is a coordinate system in which each axis corresponds to the activity of one neuron. Within this state space, each point reflects a specific pattern of neural activity in the RNN, which we refer to as a neural state. Our objective in using this framework is to characterize the mapping between task and neural state spaces. Since the number of task variables is typically much smaller than the number of neurons, numerous studies have used dimensionality reduction techniques to find a set of "latent neural dimensions" within a lower-dimensional "latent neural subspace" that more readily correspond with task variables [13,24]. Finally, we define "latent neural manifold" as all task-relevant patterns of neural activity in the latent neural space. Note that there is no guarantee of a one-to-one match between the latent neural subspace and latent task subspace, especially when the task manifold is nonlinearly embedded in the neural state space (Fig 1) [25,26].

## Engineering RNNs that embody latent task dynamics

Our central aim is to synthesize RNNs that establish desired manifolds and dynamics in the neural state space, which we refer to as the target manifold and target dynamics. We consider the class of RNNs in which the dynamics of the units are characterized by a differential equation as follows:

$$F(x) = \frac{dx}{dt} = \frac{1}{\tau}(-x + W^T \phi(x) + I) \tag{1}$$

In this equation, $x$ is an $N$-dimensional vector specifying the activity of $N$ units in the network, $\tau$ is the time constant of each unit, $W$ is an $N$-by-$N$ matrix specifying the coupling between units, and $I$ is an $N$-dimensional vector specifying the input into all units, which we assume is independent of the network state. The superscript T signifies transpose operation. The function $\phi(x)$ is a differentiable function representing the input-output transformation for each unit. Here, we use $\phi(x) = \tanh(x)$.

Our overall approach is to apply constraints on a number of "setpoints," denoted $x_j$, on the target manifold. These constraints can be applied either directly to the network dynamics (Eq 1), or to the partial derivatives, also known as Jacobians, of the RNN activity. In dynamical system analysis, Jacobians are typically used to assess local stability. Here, we exploit the relationship between Jacobians and local stability to impose constraints on network dynamics in the

vicinity of the setpoint. The Jacobian of the network, denoted $J_{RNN}$, can be written as follows:

$$J_{RNN} = \frac{\delta F_i}{\delta x_j} = \frac{1}{\tau}(-\boldsymbol{I}_N + W^T \Phi) \tag{2}$$

where $\Phi$ represents a diagonal matrix containing the derivative of $\phi(x)$ and $\boldsymbol{I}_N$ is the identity matrix of size $N$.

We sought to match $J_{RNN}$ to the Jacobian of the target dynamics, denoted $J_{targ}$. To do so, we used eigendecomposition to factorize $J_{RNN}$ to a set of eigenmodes. Each eigenmode is characterized by an eigenvector, which specifies a single dimension within the state space, and a corresponding eigenvalue that quantifies the rate and direction of movement along that dimension [29]. If we collect the $N$ eigenvectors within a matrix $U$ and the eigenvalues within a diagonal matrix $\Sigma$, $J_{RNN}$ can be decomposed to $U\Sigma U^{-1}$. We will enforce the condition that $U$ be an orthonormal matrix, so its inverse is the same as its transpose. After substituting $J_{RNN}$ with this eigendecomposition and some linear algebra, we can rewrite (2) as follows:

$$U^T \Phi W = \tau \left( \Sigma + \frac{1}{\tau} \boldsymbol{I}_N \right) U^T \tag{3}$$

Since we want $J_{RNN}$ to match $J_{targ}$, we can replace $U$ and $\Sigma$ by their corresponding values based on $J_{targ}$, and solve for $W$. The network will then be wired so that its eigendecomposition is the same as that of $J_{targ}$. However, we have to address one problem beforehand: in most cases of interest [26], the number of dimensions spanned by the target manifold, denoted $d$, is much smaller than the network dimension ($N$). From a geometrical perspective, this means that the network activity must remain within a $d$-dimensional subspace. In Eq 3, the $d$ eigenmodes of $J_{targ}$ can be used to constrain the first $d$ eigenmodes of $J_{RNN}$. For the other eigenmodes, we employ a simple trick: we set the eigenvalues of the remaining $N$-$d$ dimensions to a negative value ($1/\tau$). This serves two purposes. First, it ensures that activity along dimensions orthogonal to the manifold rapidly decays, and second, it allows us to ignore these dimensions and rewrite Eq 3 for the d eigenmodes associated with $J_{targ}$:

$$U_{targ}^T \Phi W = \tau \left( \Sigma_{targ} + \frac{1}{\tau} \boldsymbol{I}_d \right) U_{targ}^T \tag{4}$$

where $U_{targ}^T$ contains the $d$ eigenvectors of $J_{targ}$ embedded in an $N$-dimensional space ($N$-by-$d$), $\Sigma_{targ}$ contains the corresponding eigenvalues, and the identity matrix is now of size $d$. This equation can be further simplified to the following form, in which the matrices $A$ and $B$ stand in for the corresponding expressions in Eq 4.

$$A_{x_1} W = B_{x_1} \tag{5}$$

Since $J_{targ}$ is $d$-dimensional, for each setpoint on the manifold (e.g., $x_1$), Eq 5 provides $d$ linear constraints on the connectivity matrix. We can rewrite that equation for multiple setpoints on the manifold ($x_1$ to $x_m$) to create a system of linear equations to solve for the $N^2$ unknowns in $W$:

$$\begin{bmatrix} A_{x_1} \\ \dots \\ A_{x_m} \end{bmatrix} W = \begin{bmatrix} B_{x_1} \\ \dots \\ B_{x_m} \end{bmatrix} \tag{6}$$

We refer to this method as Embedding Manifolds with Population-level Jacobians (EMPJ). Using EMPJ, we can create an RNN whose activity is confined to a target manifold and whose

slow dynamics over that manifold are fully specified by the target dynamics (see Methods for full details).

## Constructing an RNN with a ring manifold and discrete fixed points

As a first test, we used EMPJ to construct a ring manifold that contains a set of discrete fixed points. This choice was motivated by the fact that (1) ring manifolds have long served as a canonical example of constrained dynamics [30], and (2) discrete fixed points can be used to introduce error-correcting dynamics over the ring as was demonstrated in a recent study of human visual working memory of color [31]. When humans report a previously seen color after a delay over a color wheel (Fig 2A, left), their reports drift slowly over the color wheel toward a stable set of colors. This behavior can be modeled by a drift-diffusion process that specifies how a latent variable (memory of the color) moves on a ring (color wheel). The model has two key components: a drift function that specifies the average movement direction and speed as a function of position on the ring (Fig 2A, middle) and the noise that causes the internal state to diffuse (see Methods).

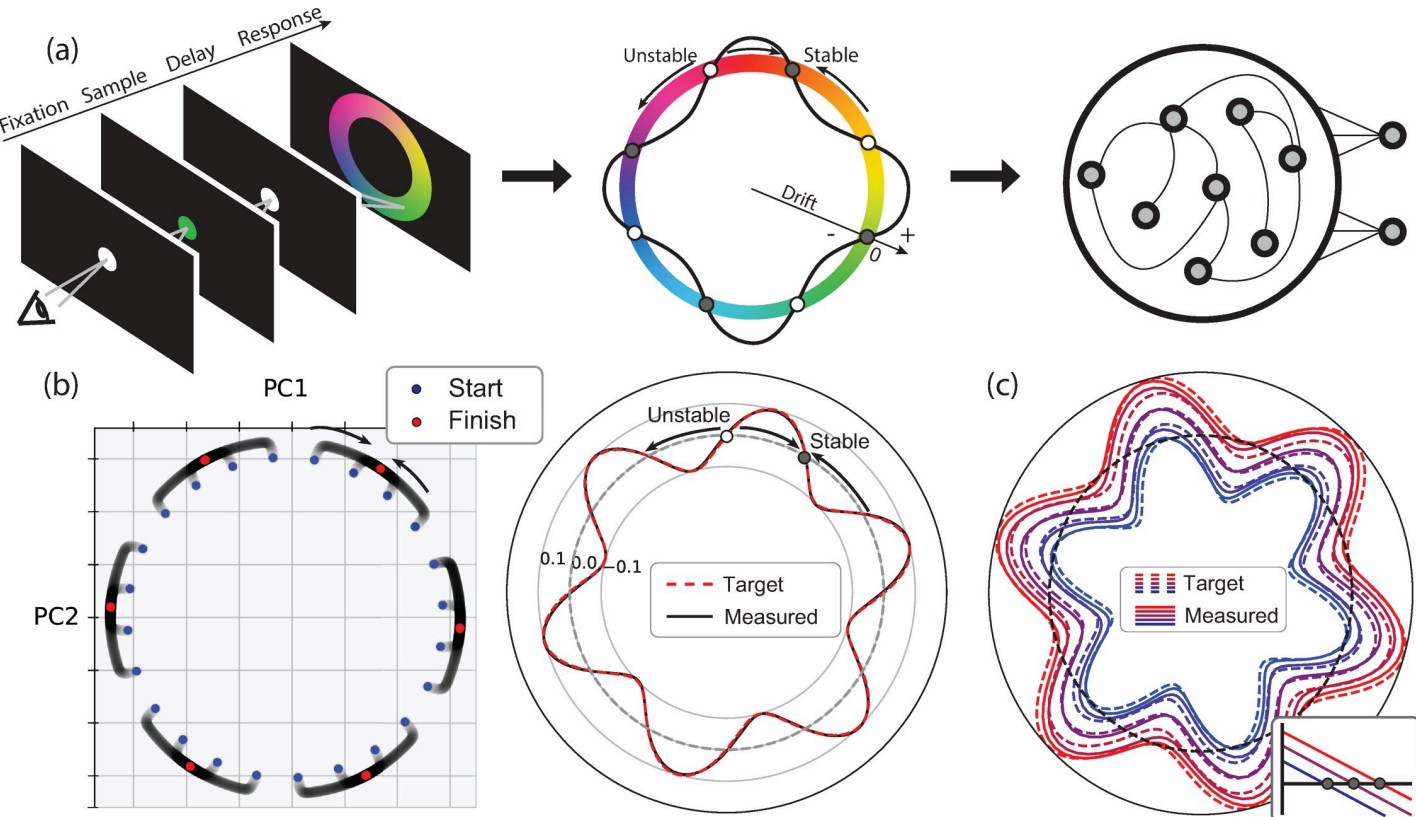

**Fig 2. Embedding a ring manifold with fixed points in a recurrent neural network.** a) Left: A color matching task. The subject must remember a sample color, and after a delay period choose the matching color on a color wheel. Middle: A hypothetical model for the color matching task. The color is encoded as a circular variable over a ring manifold. The drift function over the ring is shown by the curvy black line superposed over the ring. Clockwise and counterclockwise drifts are shown as positive- and negative-valued drifts, respectively. Wherever the drift function crosses zero, there is a fixed point which is either stable or unstable depending on the slope of the drift function at that point. Right: A recurrent neural network (RNN) model for implementing both the ring manifold and the desired drift function. b) The activity of an RNN with 400 units built to emulate a sinusoidal drift function with a period of 60 degrees. Left First two principal components of RNN activity, initialized from points close to the ring manifold (blue). Without noise, neural trajectories go towards stable fixed points (red). Right: The drift function implemented by the RNN (black) overlaid on the target drift function (dashed red). c) The drift function implemented by the RNN (solid) for five target drift functions (dashed) ranging from mostly clockwise (red) to counterclockwise (blue) motion (see Additional Constraints in Methods). Inset: illustration of zero-crossing of the target drift function near the fixed point.

We start with a simple example using a sinusoidal drift function with a period of 60 degrees so that the ring contains six equidistant and alternating stable and unstable fixed points (Fig 2B). The number of fixed points can be changed by changing the frequency of the drift function. To engineer the corresponding RNN, we first defined an arbitrary 2D embedding subspace (plane) within the neural state space that would contain the desired ring manifold. Second, we chose 64 evenly spaced positions along the ring as setpoints. Third, for every setpoint, we set the eigenvalues associated with eigenvectors orthogonal to the ring to a negative constant (see Methods for complete details). This ensures that activity near the ring would converge onto the ring. Fourth, we specified the tangential relaxation dynamics over the ring at each setpoint. To do so, we set the eigenvalues associated with the eigenvector parallel to the ring to the derivative of the sinusoidal drift function. This ensures that $J_{RNN}$ matches $J_{targ}$ at every setpoint. Finally, we solved the linear equations in Eq 6 to compute $W$, the connectivity matrix for the RNN that satisfies these constraints.

To test the solution, we initialized the network at various states close to the embedded ring in the neural state space and allowed the state to evolve according to the imposed relaxation dynamics. As expected, the network state moved onto the ring and evolved towards the nearest stable fixed points (Fig 2B, left). Moreover, the state dynamics over the ring indicated that the speed of the drift in the state space closely matched the speed predicted from the drift function (Fig 2B, right). These results generalized to ring manifolds with different numbers of fixed points, i.e., sinusoidal drift functions with different frequencies (S1A Fig).

So far, we relied on the Jacobians to synthesize RNNs. While Jacobians can powerfully constrain the derivative of the drift function, they cannot be used to set a baseline drift because derivatives are insensitive to the baseline. Indeed, with Jacobians, EMPJ can only accommodate drift functions with zero baseline (Fig 2B). Creating a drift function with a non-zero baseline requires a non-trivial adjustment to the network synthesis procedure that goes beyond the use of Jacobians. Here, we demonstrate the solution in an example where we add different baselines to a sinusoidal drift function. A non-zero baseline shifts the sinusoidal drift function up or down and thus changes the position of the zero-crossings on the ring. Therefore, one way to set a baseline is to specify the position of the zero-crossings. To do so, we took advantage of the fact that zero-crossings are the fixed points of the system and thus are the places where Eq 1 vanishes. Accordingly, we augmented Eq 1 with additional linear constraints that set Eq 1 to zero at places where the fixed points would be after adding a desired baseline. For example, adding constraints to place fixed points at the minimum of the drift function would force the entire drift function to lie above 0 (Fig 2C, red). We used this approach to engineer RNNs that establish a sinusoidal drift function with different baselines over a ring. The addition of constraints on the position of the fixed points allowed RNNs to accurately replicate the desired drift functions (Fig 2C and S1B Fig). These examples highlight the possibility of using EMPJ as a simple and rapid method for constructing RNN that can express a variety of low-dimensional latent dynamics constraints by either the Jacobians or the actual state derivatives.

## Comparison of RNN with drift-diffusion model in the presence of noise

So far, we have compared the target drift-diffusion model (DDM) and the corresponding RNN solution under noiseless conditions. To analyze the effect of noise, we can either add high-dimensional "internal noise" to individual units in the network or manifold-aligned low-dimensional "external noise" through input vectors. Previous work has found that internal noise is largely ineffective because the network is robust to noise dimensions that are not aligned to the manifold [32]. We found this to be true in our simulations: moderate levels of internal noise were largely ineffective and further increasing noise levels led to unpredictable

network behavior (S2 Fig). We therefore focused our analysis on the effect of external noise within the embedding subspace.

To compare the RNN to the DDM in the presence of noise, we first calculated the external noise levels in the RNN that would be equivalent to diffusion noise in the DDM (see Methods for details). We then simulated the RNN with external noise and compared its behavior to that of the DDM. We found that the distributions of end states on the neural manifold for a range of initial states were similar to those in the DDM. We verified these results for 2, 4, 6, 8, and infinitely many fixed points (i.e., a continuous attractor with no drift) (Fig 3A).

To quantify the performance of the RNN and DDM, we computed the average bias (BIAS), variance (VAR), and the corresponding root-mean-square-error (RMSE) of the end state distributions in the two models (see Methods for definitions). Overall, these statistics were similar across the two models for various numbers of fixed points indicating that the RNNs accurately emulated the DDMs for which they were engineered (Fig 3B and Table 1). For the specific choice of the parameters in our simulation, the BIAS was larger for fewer numbers of fixed points. This relationship is expected because of the clustering of the end states near the fixed points instead of their corresponding initial states (Fig 3A). VAR and RMSE, on the other hand, were relatively large for a ring with 2 fixed points, decreased for rings with 4 and 6 fixed points, and increased again for 8 and infinitely many fixed points. This finding suggests that RNNs with discrete stable states (i.e., ring manifolds with fixed points) may be more suitable than those with continuous attractor states for representing continuous variables. More generally, our results indicate that RNNs synthesized by EMPJ can emulate both the drift and diffusion in target DDMs.

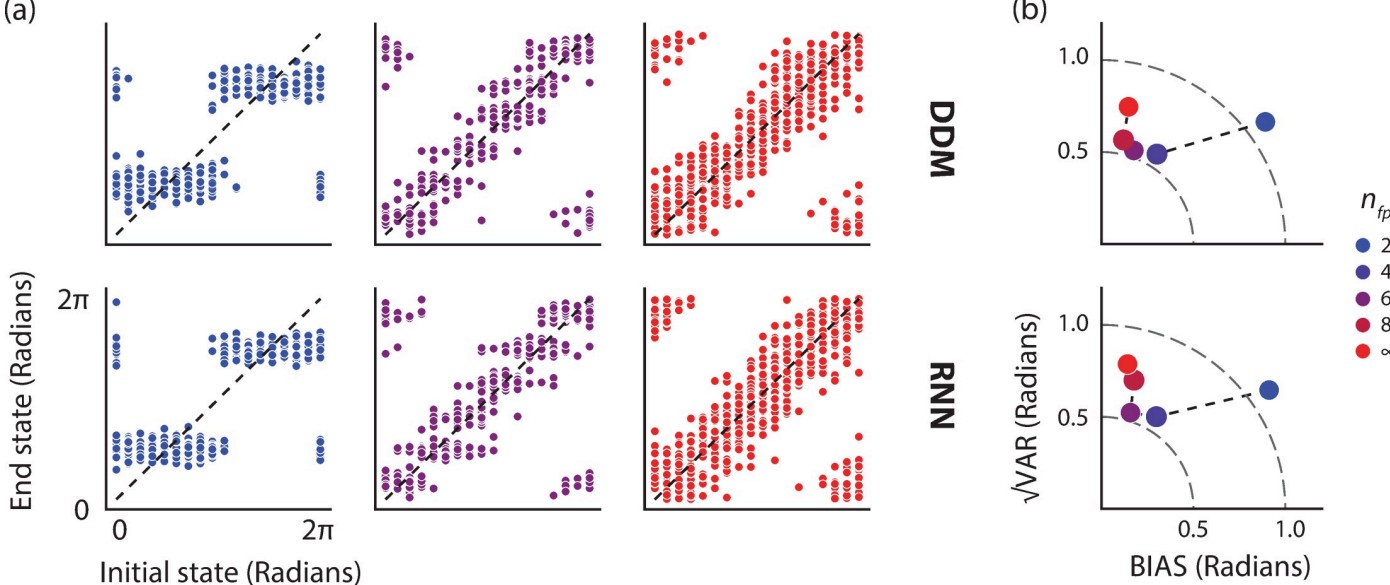

**Fig 3. Representation of a circular variable on a ring manifold with fixed points in the presence of noise.** a) Top: Initial and end state over a ring task manifold whose dynamics are governed by a drift-diffusion model (DDM) with sinusoidal drift functions creating a specified number of fixed points ($n_{fp}$). The end state for each initial state was registered after 15 sec of running the dynamics. We tested n = 18 equidistant initial states over the ring and simulated the dynamics 30 times for each initial condition. See Methods for additional details about the simulation. Bottom: Initial and end states of RNNs with 300 units engineered to replicate the task manifold and dynamics. The RNNs were simulated in the presence of external noise confined to the plane of the ring manifold (see Methods). b) Average bias (BIAS), variability ($\sqrt{VAR}$), and the corresponding root-mean-square-error (RMSE) for each simulation for the DDM (top), and the engineered RNNs (bottom). The sum of squares relationship between BIAS, $\sqrt{VAR}$, and RMSE, can be depicted on a quarter circle (dashed lines) with the radius representing constant RMSE (see Methods for definitions).

**Table 1. The average bias (BIAS), variance (VAR), and root-mean-square-error (RMSE) of the end state distributions in the target drift-diffusion model (DDM) and the corresponding recurrent neural network (RNN).** Results are shown for 2D ring manifolds with different numbers of fixed points ($n_{fp}$). For both models, the $n_{fp}$ condition associated with the lowest RMSE is underlined.

| | $n_{fp} = 2$ | 4 | 6 | 8 | $\infty$ |
|---|---|---|---|---|---|
| $\sqrt{\text{VAR}_{\text{DDM}}}$ | 0.66 | 0.49 | 0.51 | 0.57 | 0.75 |
| $\sqrt{\text{VAR}_{\text{RNN}}}$ | 0.65 | 0.5 | 0.52 | 0.7 | 0.79 |
| $\text{BIAS}_{\text{DDM}}$ | 0.89 | 0.3 | 0.18 | 0.12 | 0.15 |
| $\text{BIAS}_{\text{RNN}}$ | 0.91 | 0.3 | 0.16 | 0.18 | 0.14 |
| $\text{RMSE}_{\text{DDM}}$ | 1.11 | 0.57 | <u>0.54</u> | 0.58 | 0.76 |
| $\text{RMSE}_{\text{RNN}}$ | 1.12 | 0.58 | <u>0.55</u> | 0.72 | 0.8 |

## Input control of network dynamics

So far, we demonstrated the ability of our method to create RNNs that establish a ring manifold with various number of fixed points in the absence of any input. But what if we wish to engineer RNNs that can perform flexible computations based on contextual cues? Here, we propose a solution to this problem by building RNNs whose latent dynamics can be rapidly reconfigured by a tonic input carrying information about the context (Fig 4A), as has been suggested by modeling and physiology experiments [11–15].

The first step is similar to what we described before: we set the Jacobian for a number of set-points over a target manifold to match the target dynamics. We will refer to the subspace spanned by the associated eigenvectors as the "recurrent subspace." In the case of a ring manifold, the recurrent subspace corresponds to the 2D plane that contains the ring (Fig 4B). To

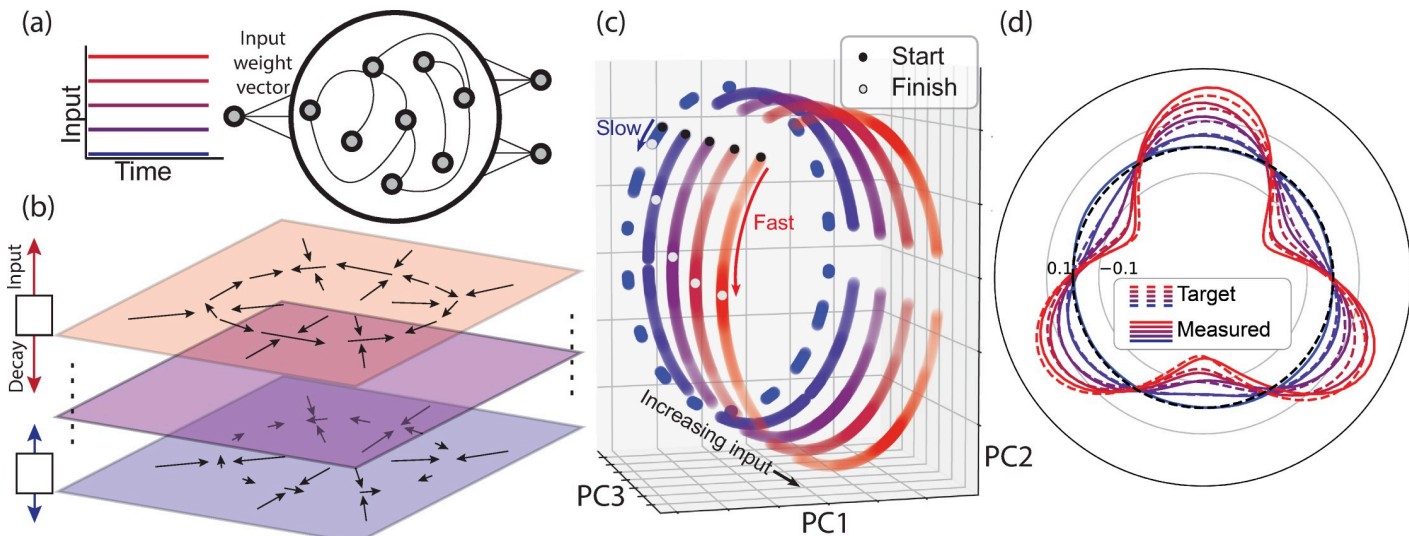

**Fig 4. Input control of speed.** a) A schematic showing an RNN whose behavior is controlled by a one-dimensional input that drives the network along an "input weight vector". b) State space representation of the ring manifold under different levels of input. The input drives the network state in the direction of the input weight vector, which is orthogonal to the plane of the ring (upward arrow). We constrain the RNN to have a negative eigenvalue along the input weight vector causing a decay in network activity (downward arrow). The balance between input and decay determines the final position of the plane containing the ring. In the schematic shown, for all input levels, the network contains a ring manifold (radial arrows converging onto a circle), but the drift is faster (longer tangential arrows) for stronger input (red). c) First three principal components (PC1-PC3) of network activity, initialized at various points around the ring and for different input conditions. Black and white dots illustrate the initial and terminal states for an example simulation for each input level. As expected, tonic inputs confine the dynamics to different rings. (d) Measured (solid) and target (dashed) drift function for various input levels shown with different colors (blue: weak input; red: strong input; tested values: 0, 0.5, 1, 1.5, and 2).

exert input control, we chose a subspace orthogonal to the recurrent subspace, which we refer to as the "input subspace." We set the eigenvalues associated with the dimensions of the input subspace to be a negative constant so that, in the absence of inputs, the state along all dimensions of the input subspace will decay to zero (see Methods for details). However, if we drive the network with a tonic input along a specific dimension of the input subspace, the interaction between the input and the decay will cause the system to settle at a non-zero equilibrium point along that dimension. In other words, the input will shift the recurrent subspace along the dimensions of the input. Our strategy for engineering an RNN whose dynamics can be adjusted by the input is to specify different dynamics (i.e., different Jacobian) for different input levels (see Methods).

Here, as an example, we focused on a simple case in which we required the RNN to establish a ring manifold, and required that the speed of the drift (i.e., the amplitude of the sinusoid) over the ring be adjusted by a tonic input. For this example, the recurrent space is the plane that contains the ring (two-dimensional), and the input subspace is a scalar (one-dimensional). We synthesized this RNN using the following step-by-step procedure: 1) We chose a pair of orthogonal eigenvectors to define the plane of the ring. We refer to these as planar eigenvectors. 2) We chose an orthogonal eigenvector for the input. 3) We chose $M$ input levels. 4) We found the equilibrium point of the plane along the input dimension for each of the $M$ input levels. This defined $M$ parallel planes, one for each input level. 5) We chose setpoints on each of the $M$ parallel planes. 6) At each setpoint, we set the eigenvalues associated with the planar eigenvectors such that the local drift over each plane was proportional to the input associated with that plane.

The resulting RNN was able to establish the desired input-dependent dynamics: there was nearly no drift in the absence of input, and progressively faster drifts for stronger inputs (Fig 4C and 4D). While this example focused on a one-dimensional input, with multidimensional inputs, one should be able to synthesize more sophisticated systems that afford parametric control of the latent dynamics in the recurrent subspace.

## Rings embedded in high-dimensional space

So far, we focused our work on perfectly circular ring manifolds that reside in a two-dimensional latent space. An implicit assumption in this simple implementation is that the entire variance of the signal across the population resides in two dimensions. However, in biological networks, signals can span many more dimensions even when the latent manifold is a one-dimensional ring [32]. Therefore, we used our approach to examine the properties of RNNs for which the ring is embedded in higher dimensions (i.e., high-dimensional ring). To facilitate the presentation of the results, we will use a notation of the form $O(n_{dim}, n_{fp})$ to refer to a ring manifold embedded in $n_{dim}$ dimensions with $n_{fp}$ fixed points. For example, $O(2,4)$ corresponds to a planar ring (2D) with 4 fixed points.

To construct such RNNs, we need to specify two properties: the geometry of the ring and the dynamics over it. To specify the geometry, we parameterized the ring in terms of a set of circular functions. For the simple case of a circular two-dimensional ring, we need two latent tuning functions, a sine and a cosine of the same frequency (Fig 2). Since the activity of the RNN is confined to the ring, it is expected that the tuning function of all units in the RNN is a weighted sum of these two latent tuning functions. At the other extreme, we can have a circuitous ring that visits all dimensions [30]. In this case, we expect units in the network to have independent tuning functions (Fig 5A, left). More generally, to create a ring manifold that resides in a latent space of dimension $K$ (Fig 5B), the units' tuning functions should be a weighted sum of $K$ independent tuning functions (Fig 5A, right).

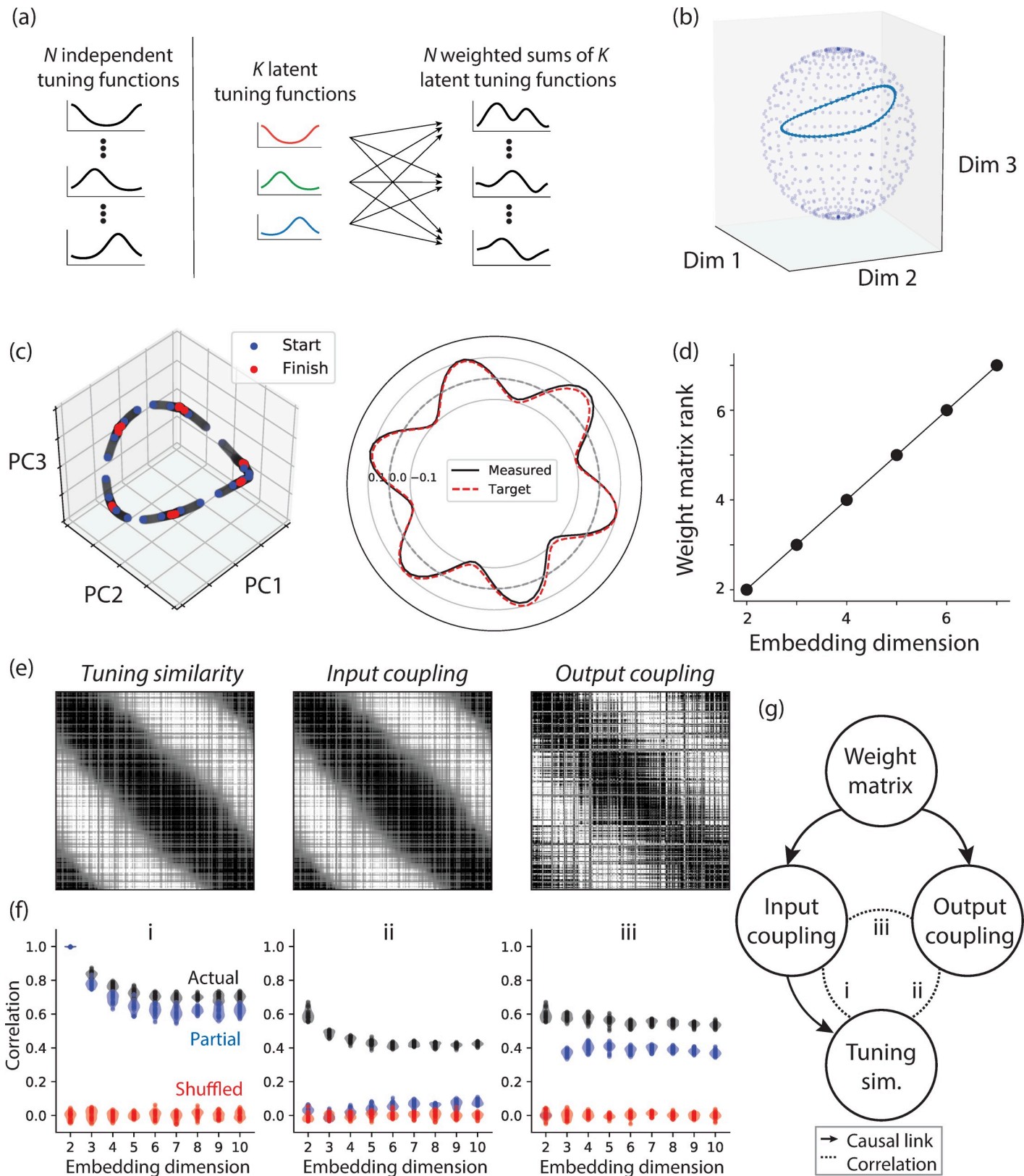

**Fig 5. Embedding rings in higher dimensions.** a) An illustration of RNN units in terms of their tuning function for a network with an embedded ring manifold. The left column illustrates a case in which the *N* units in the RNN have independent tuning functions. The right column illustrates a case in which the *N* tuning functions

are weighted sums of $K<<N$ latent tuning functions (the plot corresponds to 3 latent tuning functions shown in color). b) An example ring manifold embedded in 8 dimensions shown over a 3D slice of a hypersphere. The first two dimensions (Dim 1 and Dim 2) correspond to the sinusoidal tuning functions, and the third dimension (Dim 3) corresponds to one of the latent tuning dimensions. In this example, the ring was made from 6 von Mises functions and 2 sinusoidal functions. c) Left: The initial state (blue), intermediate states (gray), and end states (red) of an RNN engineered to have O(8,6), plotted in a subspace spanned by the first three principal components. Right: The target drift function and the drift function measured from the engineered RNN. d) The rank of the RNN weight matrix, determined by the number of non-zero eigenvalues, matches the ring's embedding dimension. e) Left: The tuning similarity matrix measured as the covariance of tuning functions for an RNN engineered to have O(2,6). Middle: The input coupling matrix measured as the inner product of the input weight vectors for all pairs of units for the same RNN. Right: The output coupling matrix measured as the inner product of the output weight vectors for all pairs of units for the same RNN. Matrix indices are ordered by the preferred orientation of the tuning curves. f) Left: Violin plot of the correlation coefficient between the elements of the tuning similarity and input coupling matrices of engineered RNNs. The results are shown for various ring embedding dimensions with no drift over the ring. The actual correlations are shown in black, partial correlations controlling for the output coupling matrix in blue, and the values expected by chance in red. To compute chance levels, we shuffled the label of tuning functions (i.e., the rows of the tuning similarity matrix) and recomputed the correlations. Middle: Same analysis repeated for the tuning similarity and output coupling matrices, with the partial correlations controlling for the input coupling matrix. Right: Same analysis repeated for the input coupling and output coupling matrices, with the partial correlations controlling for the tuning similarity matrix. g) Causal graph showing the relationship between the input coupling matrix (left node), output coupling matrix (right node), tuning similarity matrix (bottom node), and the weight matrix (top node). The three dashed lines marked as i, ii, and iii correspond to the three correlation plots in panel f (left to right).

The exact geometry of the ring depends on the profile of the latent tuning functions. In our simulations, we used $n_{dim}$ tuning functions to construct a ring of dimension $n_{dim}$. Two of the tuning functions were a pair of sine and cosine of the same frequency creating a planar ring. The other dimensions that were associated with excursions away from the plane were parameterized as $n_{dim}$-2 evenly-spaced von Mises functions with width parameter, $\kappa$ (S3A Fig). Additionally, at every point along the ring, we adjusted the amplitude of the sine and cosine functions so that the ring would reside on the surface of a hypersphere (S3B Fig; see Methods for details).

We used EMPJ to construct the RNN to establish the high-dimensional ring associated with the latent tuning functions. Simulations indicated that the RNN states were confined to the ring and their corresponding dynamics matched the target dynamics (Fig 5C), although the overall stability was lower for higher-dimensional rings (S4 Fig). As expected by the weighted sum operation (Fig 5A), the tuning function of individual units in the resulting RNN was heterogeneous and complex (S3C Fig). The ring length also varied with both embedding dimension and the width parameter of the von Mises tuning functions (S3D Fig).

We reverse-engineered the RNNs to investigate the properties of the underlying connectivity matrix. The rank of the connectivity matrix was the same as the number of embedding dimensions, independent of the drift function (Fig 5D). This is expected because the system of linear equations that we used to compute the connectivity matrix was constrained by data points over the ring.

In canonical ring attractor models, the strength of the connectivity between units reflects the distance between their "preferred" angles (i.e., the angle associated with peak activity) [30,33]. Such structure has also been observed in trained RNNs [34]. Here, we asked whether RNNs engineered using EMPJ exhibit a similar characteristic. To do so, we analyzed the relationship between three similarity matrices: (1) the similarity between tuning functions of different units, which we refer to as the *tuning similarity matrix*, (2) the alignment between the input weight vectors for all units, which we refer to as the *input coupling matrix*, and (3) the alignment between output weight vectors for all units, which we refer to as the *output coupling matrix*. For the tuning similarity matrix, we used the covariance between the tuning functions so that element $(i, j)$ contains the covariance between the tuning functions of the units $i$ and $j$. For the input coupling matrix, we used the product of the connectivity matrix $W$ with its transpose $W^T$ so that element $(i, j)$ contains the inner product of the input weights into unit $i$ and $j$. For the output coupling matrix, we used the product of $W^T$ with $W$, so that element $(i, j)$ contains the inner product of the output weights of units $i$ and $j$. Finally, we measured element-wise correlations between these matrices to quantify the degree to which tuning functions

(first matrix), input couplings (second matrix), and the output couplings (third matrix) were similarly structured.

We first applied this analysis to the simplest network with a 2D ring manifold, i.e., the canonical ring attractor. In this case, the tuning similarity matrix (Fig 5E, left), the input coupling matrix (Fig 5E, middle), and the output coupling matrix (Fig 5E, right) had a circular structure, and were strongly correlated with one another (Fig 5F). These relationships mimic classical models of ring attractors, which rely on a symmetric weight matrix with perfect correlations between tunings, input couplings and output couplings [30,33]. This finding suggests that the solution EMPJ finds is similar to classical ring models even though EMPJ does not explicitly require the weight matrix to exhibit symmetry.

Next, we used the same analysis to evaluate networks with higher dimensional ring manifolds. All pairwise correlations were strong (Fig 5F, black) relative to the null computed from shuffling the elements (Fig 5F, red). However, the correlation between the tunings and both input and output couplings dropped gradually for higher dimensional manifolds (Fig 5F, left and middle), which is expected given that higher dimensions lead to deviations from the ring structure.

To better understand the nature of these three-way relationships, we considered the causal graph that captures the relationship between the tuning functions, input couplings, and output couplings (Fig 5G). Both the input and output coupling matrices depend solely on the weight matrix $W$ (Fig 5G, top solid arrows). This is consistent with the robust correlations between the input and output couplings, which suggest that weight matrices for higher dimensions retain some symmetry (Fig 5F, right). Since the unit responses are driven by their inputs, we also expect the input coupling to causally control the tuning similarity (Fig 5G, bottom solid arrow). Accordingly, the tunings and input couplings had the strongest correlations irrespective of dimensionality (Fig 5F, left). Finally, the causal graph also indicates that the relationship between the tunings and output couplings (Fig 5G, marked ii) is mediated by the input couplings. We tested this prediction by measuring the partial correlation between the tunings and output couplings while accounting for the input couplings. The correlations dropped precipitously (Fig 5G, middle, blue), which is expected given the underlying causal structure between these variables.

## Limitations of RNN dynamic capacity

Higher dimensional rings may have advantageous coding capacity as they can increase discriminability between nearby states (S3D Fig). However, this increase in dimensionality may come at the cost of stability. Intuitively, higher dimensional rings consist of regions with higher curvature requiring faster changing gradients that may be more difficult to implement in an RNN. To assess this possibility quantitatively, we analyzed the dynamical stability of RNNs associated with high-dimensional rings. To do so, we must first define a suitable error metric. Measuring the mean squared error between the instantaneously measured drift along the ring and the target drift function will not suffice, since trajectories may fly off the ring. We therefore introduced a metric referred to as "deviation," illustrated in Fig 6A. At various points in time for an RNN trajectory, we decoded the current angle represented on the ring, and then calculated what the RNN state should be given that value. We used the Euclidean distance between the RNN's actual state and where it should be as our measure of deviation at that point in time. Finally, to get an overall measure of stability for a specific RNN, we calculated mean deviation, averaged over time and initial conditions.

Analysis of the mean deviation revealed several properties of dynamical stability in networks featuring high-dimensional rings. First, larger networks are generally more stable for

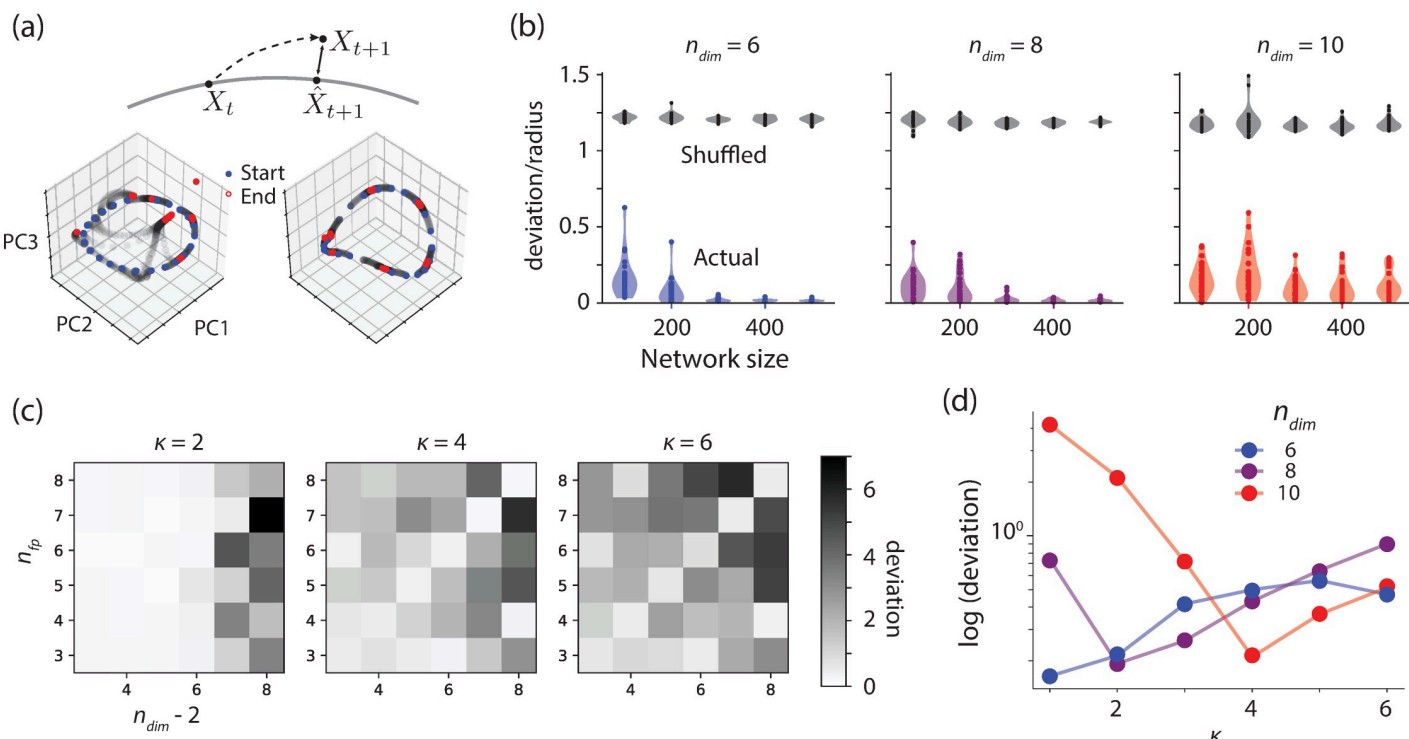

**Fig 6. Constraints on network performance.** a) Top: Illustration of the deviation metric used to quantify network performance. The deviation is measured as the Euclidean distance between the actual neural state and its corresponding state on the ring that would lead to the same decoded angle (see Methods). Bottom: The initial state (blue), intermediate states (gray), and end states (red) of two networks engineered to have O(6,6), plotted in a subspace spanned by the first three principal components. The deviation in the left panel is larger because the end states are farther from the target ring manifold. b) Deviation relative to ring radius as a function of network size for RNNs engineered to have O(6,4). Each dot represents one network. The colored symbols correspond to the actual deviation values. The gray symbols represent the ceiling values for deviation that are expected from a network whose states have no lawful relationship to states on the ring manifold. To compute ceiling deviations, we measured the Euclidean distance between the actual neural state and randomly chosen states over the ring manifold. c) Deviation for RNNs engineered to have O($n_{dim}$,$n_{fp}$) for different values of the width parameter of the von Mises tuning function ($\kappa$). Higher values of $\kappa$ indicate narrower latent tuning curves. d) Logarithm of deviation as a function of $\kappa$ for RNNs engineered to have ring manifolds with the same number of embedding dimensions and fixed points.

rings with different numbers of fixed points and embedding dimensions (Fig 6B). Second, the geometry of the ring impacted its dynamical stability. In our formulation, the geometry was fully determined by the number of the underlying von Mises tuning functions ($n_{dim}$-2) and their tuning width ($\kappa$). We therefore quantified mean deviation as a function of these variables. Results indicated the networks were more stable when the number of fixed points ($n_{fp}$) matched the number of von Mises tuning functions, especially when those tuning functions were narrower (Fig 6C). Moreover, for conditions in which $n_{fp}$ matched $n_{dim}$-2, the value of $\kappa$ associated with lowest mean deviation increased lawfully with embedding dimensionality (Fig 6D). In sum, our analyses reveal nontrivial relationships between the geometry and dynamical stability of high-dimensional rings established by RNNs. We find that high-dimensional ring manifolds can accommodate a larger number of stable fixed points if the number of fixed points matches the number of the underlying tuning functions and if the tuning functions have an appropriate width.

## Discussion

We have developed a method, EMPJ, for synthesizing RNNs that perform computations by implementing specific task-relevant dynamics. EMPJ works by specifying local constraints on the dynamics, resulting in the desired global behavior. The key innovation in EMPJ is that it

derives the network connectivity directly from a set of linear equations given by those constraints. We demonstrated the utility of this technique in the context of a simple working memory task in which the network dynamics were specified by a drift diffusion process over a ring-shaped manifold. The flexibility of EMPJ enabled us to implement a variety of drift functions over the ring accurately. For example, we were able to create networks whose dynamics established drift functions with error-correcting properties in the presence of noise.

Moreover, we used EMPJ to generate networks whose dynamics can be flexibly adjusted by an input. This opens the possibility of creating models of neural systems that perform context-dependent sensorimotor and cognitive computations. We used this approach to model how thalamo-cortical inputs might adjust the speed with which cortical dynamics evolve, as has been suggested by recent findings [11,35]. However, unlike end-to-end training methods [11], EMPJ enabled us to straightforwardly synthesize RNNs in which an input drove the system to different regions of state space with different drift functions. Although we focused on simple control via tonic inputs, future work should be able to extend EMPJ to incorporate richer time-varying inputs, such as pulses or oscillations, to accommodate more sophisticated control mechanisms.

One question that deserves further consideration is how to choose appropriate target dynamics for the network. In our case, we were able to engineer the target dynamics based on the computational demands of the task we considered. In general, it might be difficult to engineer such simple solutions for complex tasks whose computations involve higher-dimensional manifolds. This problem may be solved by integrating our method with other techniques that furnish the target dynamics. One option would be to use Jacobians estimated from neural spiking data recorded from an animal trained to solve the task [36]. Another option is to take Jacobians from an auxiliary artificial neural network that contains task-relevant dynamics [37]. These methods would generate target dynamics from a system able to solve the task, which could then be used with EMPJ to directly engineer an RNN with those dynamics.

Using EMPJ to synthesize RNNs with richer dynamics will require expanding its dynamical repertoire beyond the use of real eigenvalues and orthogonal eigenvectors. For example, to introduce oscillatory dynamics, we need to extend EMPJ to handle complex eigenvalues. Another improvement is to engineer networks with constraints that are complementary to Jacobians. Jacobians are useful because they afford specification of desired stability criteria around a setpoint and can be used to implement input control (Fig 4). However, Jacobians are not suitable for sculpting regions of the state space where state derivatives are not well-behaved (e.g., are too small). For example, Jacobians cannot be used to set drift functions with non-zero baselines or create constant vector fields in the state space. However, as we demonstrated for the case of non-zero baseline (Fig 2C), this problem can be addressed by imposing additional constraints on the dynamics (Eq 1). Therefore, another area of development is to explore the opportunities afforded by combining Jacobians with other constraints such as those that can be placed directly on state dynamics.

As described, EMPJ provides the means for embedding a task parameter manifold directly into an RNN. The approach is similar to that described by the Neural Engineering Framework (NEF), which also matches latent task dimensions to latent neural dimensions and creates a recurrent weight matrix that produces the desired transformations of neural representations [21]. One point of contrast is that EMPJ only requires knowing local linear approximations of dynamics, while the NEF involves specifying the global dynamics equations. Therefore, EMPJ can be more effective when the global equations are unknown, but only if local linear approximations are sufficiently constraining. Moreover, in networks constructed by EMPJ, off-manifold perturbations decay rapidly because eigenvalues associated with off-manifold eigenvectors are explicitly set to negative values. The NEF, on the other hand, does not use Jacobians, and therefore, local stability over the manifold is not explicitly imposed. Our approach also makes

it easier to create networks for which the latent task manifold is embedded nonlinearly in the neural manifold. Given these differences, we present EMPJ as a complementary technique to the NEF, as it shares the same underlying principles.

EMPJ can also be contrasted with other RNN synthesis methods. For example, one might test the degree to which the connectivity matrix resulting from EMPJ matches predictions from other approaches that relate connectivity to low-dimensional dynamics. Two recent examples of such work are based on mean field theory [22] and distributions of network motifs [38]. Generally, the connectivity matrices found through EMPJ may be different from those found through mean-field methods. This is possibly because mean-field methods rely on the properties of the distribution from which the connectivity matrix weights are drawn, while the weights found through EMPJ are less constrained.

A larger goal of analyzing and synthesizing RNNs is to gain a deeper understanding of the relationship between manifold geometry, complexity of dynamics, and network characteristics. EMPJ makes it easy to generate and test hypotheses about those properties. For example, we used EMPJ to assess how the embedding dimensionality of the manifold and the organization of fixed points impact the ease of implementing different drift functions. Future directions could extend this work to further investigate general properties of network models such as capacity [39] and manifold smoothness [40].

## Methods

In the main manuscript, we explained the general procedure for using EMPJ. Here, we focus on a few design considerations that can help EMPJ function more smoothly.

### Choice of setpoints

The first step in EMPJ is to choose setpoints on a target manifold. The exact number of setpoints does not matter, but the sampling should be sufficiently dense so that the system can interpolate the drift function between points. For the ring manifold example focused on in this paper, the number of setpoints can be small. For example, 12 setpoints is enough to engineer an RNN to have a 2D ring with 6 stable fixed points (S5 Fig). However, we typically used at least 64 setpoints to be safe.

### Choice of input-output nonlinearity

The choice of input-output nonlinearity we impose on the RNN units can limit the space of possible target dynamics that the network can implement. In our case, we chose the hyperbolic tangent function for all units, which is an odd function (i.e., tanh(-x) = -tanh(x)). This choice forces the vector field gradients at opposite points relative to the origin to be equal with opposite signs. As a corollary, if the manifold is symmetric about the origin, then one can limit the setpoints to only one half of the manifold. One consequence of this symmetry is that a 2D ring manifold with sinusoidal drift function must have an even number of fixed points. To make rings with odd numbers of fixed points, some kind of asymmetry must be introduced. For example, this may be achieved by considering a heterogeneous library of input-output nonlinearities for the units in the network, or requiring that the manifold is not symmetric with respect to the origin (e.g., high dimensional ring with a non-zero origin).

### Choice of embedding

Generally, the speed at which neural states of an RNN evolve depends on the units that operate near their saturating nonlinearity [11,41]. If all units were nearly saturated, the network could

only evolve at very slow speeds. It is therefore important to set the RNN up so that the operating regime is compatible with the speed requirements of the desired drift function. In our framework, this can be achieved by an appropriate choice of the region of the neural state space in which the target manifold is embedded. Generally, points farther away from the origin of the neural state space are associated with some units operating closer to the saturation points. For a ring manifold centered on the origin, increasing the radius of the ring (i.e., stronger activations) would drive units closer to their saturating nonlinearity, and thus reduce the speed. For the relatively slow drift functions we considered in this manuscript, it was important to have some of the units be near their saturation point (S6 Fig).

## Solving the linear equations

To ensure that the RNN solution is not the result of a specific choice of setpoints, embedding or other specifics, it is advantageous to regularize the solutions by adding noise to the linear equations as follows:

$$(A + \xi)W = B \tag{7}$$

$\xi$ denotes a matrix of white noise ($\sigma = 10^{-6}$ in all cases unless noted otherwise) the same size as $A$, which serves as a regularizer to create more robust solutions. We used the least-squares solver from the NumPy linear algebra library to solve the linear equations.

## Ring manifold with fixed points

For our initial example (Figs 2 and 3), we focused on a 2D ring manifold and set the dynamics according to a sinusoidal drift function of the form $G(\theta) = -0.1\cos(\omega\theta)$ without any diffusion (i.e., no noise). In this formulation, the frequency $\omega$ determines the number of stable fixed points around the ring. The corresponding Jacobian ($J_{targ}$) at angle $\theta$ is $(\theta) = 0.1\omega\sin(\omega\theta)$, which is the derivative of the drift function at that point.

To engineer the corresponding RNN, we chose 64 evenly spaced setpoints around the ring ($\theta_i = 2\pi i/64$), and embedded them along the perimeter of a planar ring centered at the origin of a 400-dimensional neural state space representing an RNN with 400 units. The plane containing the ring was chosen arbitrarily using two orthogonal projection vectors. We set the radius of the ring to 10 so that a subset of units would operate near their saturation level. For each $\theta_i$, we set the eigenvalue associated with the tangential eigenvector (i.e., the vector within the plane that is tangent to the ring) to $(\theta_i)$.

## Additional constraints on the RNN equation

We focused on engineering RNNs using Jacobians, which afford specification of desired stability criteria around each setpoint. However, because Jacobians are related to the derivative of the drift function, they cannot be used to specify baseline drifts other than zero. This apparent limitation can be straightforwardly addressed if we augment the linear equations in EMPJ with additional constraints placed directly on the RNN equation instead of its Jacobian. To set the baseline, we took advantage of the fact that zero-crossings of the drift function represent fixed point states ($x_f$) at which we expect $dx/dt$ to vanish. Setting Eq 1 equal to zero at the desired zero-crossing points, we can write:

$$x_f = W^T\phi(x_f) \tag{8}$$

Adding constraints based on the location of fixed points to Eq 6, we can rewrite the full set of linear equations as follows:

$$\begin{bmatrix} A_{x_1} \\ \cdots \\ A_{x_m} \\ \phi(x_f) \end{bmatrix} W = \begin{bmatrix} B_{x_1} \\ \cdots \\ B_{x_m} \\ x_f \end{bmatrix} \tag{9}$$

In Fig 2C, we tested baseline values of -0.1, -0.07, 0, 0.07, and 0.1 by shifting the location of the zero-crossings in increments of one eighth of the drift function's period.

## Simulations

After engineering the RNN, we tested its behavior using simulations, typically starting from initializations evenly spaced around the ring. We measured the performance of the network in terms of the difference between the expected angle, $\theta$, based on the target drift function, and inferred angle, $\hat{\theta}$, based on the state over the ring in the RNN. To infer $\hat{\theta}$, we constructed a simple $Nx2$ linear decoder, $D$, that mapped the N-dimensional state of the network to the cosine and sine of $\hat{\theta}$ as follows:

$$\begin{bmatrix} \cos\hat{\theta} \\ \sin\hat{\theta} \end{bmatrix} = D \tanh(x) \tag{10}$$

$$\hat{\theta} = \tan^{-1}\left(\frac{\sin\hat{\theta}}{\cos\hat{\theta}}\right) \tag{11}$$

Note that for a planar ring, it is possible to compute $\hat{\theta}$ analytically using the projection vectors and the network's recurrent weight matrix. However, we used this linear decoding strategy as it applies more generally to rings with nonlinear embeddings in higher dimensions.

## Simulations in the presence of noise (diffusion)

In the full drift-diffusion model (DDM), the dynamics of the angle around the ring, $\theta$, is determined by a following stochastic ordinary differential equation:

$$d\theta = G(\theta)dt + \sigma_{DDM}dW \tag{12}$$

where $G(\theta)$ is the deterministic drift function and $dW$ represents a Wiener process that introduces Gaussian noise at every timestep, scaled by the standard deviation, $\sigma_{DDM}$. We note that the diffusion term must be scaled with $\sqrt{dt}$ so that the variance of the process grows with time.

For the simulations, we used a sinusoidal drift function with an amplitude of 0.2 rad/s and set the standard deviation of noise, $\sigma_{DDM}$, to 0.2. The frequency determined the number of fixed points of the drift function, and we tested values of 0, 2, 4, 6, and 8. Setting the frequency to 0 results in no drift and creates a continuous attractor over the ring. We simulated the two models 30 times each for 18 different initial conditions. The timestep was set to 50 ms for the DDM, and each trial was simulated for 15 seconds.

We also simulated the engineered RNNs in the presence of external noise. We added external noise by multiplying each projection vector used to construct the ring by a sample from a Gaussian distribution, which we then added to the network activity. This way, noise was confined to the same plane as the ring. We used the same noise level as the DDM model ($\sigma_{DDM} = $

*0.2*). However, because the effect of noise in the RNN accumulates with both the number of timesteps relative to the network's time constant (longer simulations) and the length of the ring (larger distances), we scaled the noise amplitude by $\tau/\sqrt{dt}$ and by the ring's radius, *r*:

$$\sigma_{RNN} = \sigma_{DDM} \frac{r\tau}{\sqrt{dt}} \tag{13}$$

## Bias/variance comparison

We quantified the performance of the engineered RNN by comparing the actual end state on the ring manifold (based on simulations) to the desired end state (based on the target dynamics) using three statistics [42]: BIAS, $\sqrt{VAR}$, and RMSE. BIAS summarizes the difference between actual and desired end states across all initial states tested (indexed by *i* going from *1* to *N*). $\sqrt{VAR}$ summarizes variability of end states around the average end state across all initial states tested. If we denote the bias and variance for each initial state by $bias_i$ and $var_i$, respectively, BIAS and $\sqrt{VAR}$ can be computed as follows:

$$BIAS = \sqrt{\frac{1}{N} \sum_{i=1}^{N} bias_i^2} \tag{14}$$

$$\sqrt{VAR} = \sqrt{\frac{1}{N} \sum_{i=1}^{N} var_i} \tag{15}$$

## Input control

We use recurrent subspace to refer to the subspace to which the network activity is confined in the absence of any input. For input control, we use a subspace orthogonal to the recurrent subspace, which we refer to as the input subspace. To explain our methodology, it is useful to focus on the simplest case of a one-dimensional (scalar) input. In this case, the input subspace reduces to a single dimension. To achieve input control, we choose the input direction to be orthogonal to the recurrent subspace and assign a negative eigenvalue ($\lambda$) to it so that dynamics along the input dimension will decay exponentially. This decay ensures that the behavior of the RNN does not change in the absence of input. However, if we drive the network with a tonic input, *I*, along this dimension, the interaction between *I* and the decay will cause the system to settle at a non-zero state along the input dimension. The stable state in the presence of *I* is the solution of the differential equation $0 = dy/dt = -\lambda y + I$, which is $I/\lambda$.

We used this property to our advantage by specifying different target dynamics in the recurrent subspace for different levels of tonic input (i.e., for different positions along the input dimension). This allowed us to create an array of drift functions, one for each level of input so that the input could choose the appropriate dynamics on the manifold.

In the example of the ring manifold, we added input control to create a cylinder-shaped manifold, and increased the drift speed over the ring as a function of the input strength. The functional relationship between the drift speed and the tonic input can be chosen arbitrarily. For our simulations, the ring radius was set to 8 and $\lambda_i$ was set to -1. The input levels were 6 units of distance apart, and the drift speed increased linearly with input level. In Fig 4, we used input levels of 0, 0.5, 1, 1.5, and 2.

## Constructing high-dimensional rings

We constructed high-dimensional rings in terms of a set of periodic functions over $\theta$, which we refer to as latent tuning functions. To facilitate parameterization, we used sine and cosine

to define the first two tuning functions (as for the planar ring) and von Mises functions for all additional dimensions (S3A Fig). The von Mises functions had an amplitude of 0.5, a width parameter, $\kappa$, and were evenly spaced around the ring. The equation for the $j$th tuning function ($j > 2$) for a ring with excursions in $d$ additional dimensions (i.e., $d$ von Mises functions) can be written as:

$$c_j(\theta) = \frac{1}{2} e^{\kappa(cos(\theta - \frac{2\pi j}{d}) - 1)} \tag{16}$$

We applied a scaling factor to the sine and cosine to ensure that every point of the high-dimensional ring lies on the surface of a hypersphere (S3B Fig). The radius we used in our simulations was 12, unless specified otherwise. With this radius, only a few of the units in the network operated near their saturation point.

To facilitate the presentation of the results, throughout the manuscript, we use the notation $O(n_{dim}, n_{fp})$ to refer to a ring manifold embedded in $n_{dim}$ dimensions with $n_{fp}$ fixed points.

### Ring capacity

To measure the ability of a network to approximate dynamics over a given ring, we defined an error metric we refer to as deviation. We defined deviation as the average Euclidean distance between the network state $x$ and the network state $\hat{x}$ we would expect based on the decoded angle $\hat{\theta}$, averaged over $T$ timepoints across $N$ simulations with random initializations, as follows:

$$deviation = \sqrt{\frac{1}{NT} \sum_{t=0}^{T} \sum_{i=1}^{N} (x_t^i - \hat{x}_t^i)^2} \tag{17}$$

The first step for computing deviation is to compute $\hat{\theta}$. We did this using the same procedure described in Eq 10 and Eq 11. We then used the known latent tuning functions to generate a network state $\hat{x}$ corresponding to that angle. For our measurements of deviation, we did this for 24 different initial conditions on the ring and for 5 seconds of simulation time, sampling the trajectories every 0.1 seconds unless specified otherwise. It is worth noting that the exact value of deviation is not necessarily meaningful, but it is useful for comparing different networks.

When measuring network capacity as a function of network size, we measured the deviation for 30 different networks. For each, we set the tuning function width to 2 and the number of fixed points to 4. Another relevant parameter was the standard deviation of regularization noise added when finding the weight matrix, which we set to 1e-3. We tested network sizes of 100, 200, 300, 400, and 500 units with ring embedding dimensions of 6, 8, and 10.

For measuring network capacity as a function of the ring's embedding dimensionality, width of latent tuning functions, and number of fixed points, we used a similar procedure, this time keeping the network size fixed at 400 units and changing only the parameters of interest.

## Supporting information

**S1 Fig. Comparison of the target and measured drift functions.** a) Drift values around a planar ring with different numbers of fixed points (unity line: dashed). b) Same as (a) for a planar ring with 6 fixed points and a drift function with difference baseline values. Results indicate that the measured drift values closely follow the target values.
(PDF)

**S2 Fig. Performance with respect to internal noise.** To test how the network responds to noise, we added independent Gaussian noise to each unit of a 400-unit RNN at every timestep for 5 seconds. The RNN was constructed for O(2,6). a) The input-output mapping of each unit with 4 levels of noise. The standard deviation of noise ($\sigma$) is shown on top. The shaded region shows the expected shift in the output for signals deviating one standard deviation from the mean. b) The initial state (blue), intermediate state (black), and end states (red) of network activity plotted in a subspace spanned by the first three principal components for 20 evenly spaced initializations around the ring. For low-noise regimes, the network converges on the desired fixed points and stays confined to the ring. Intermediate noise causes the trajectories to diffuse slightly but remain close to the fixed points. For the highest noise, trajectories are no longer confined to the fixed points. Attempts at mapping the level of network noise to noise in a corresponding drift-diffusion model were not successful. c) A circular histogram of decoded end states around the ring after 5 seconds across 100 evenly spaced initializations. The six fixed points are located every 60 degrees.
(PDF)

**S3 Fig. High-dimensional rings on hyperspheres.** a) Example of latent tuning functions for a hypersphere in an 8-dimensional subspace (2 sinusoidal and 6 von Mises functions). b) Due to normalization across latent tuning functions, the ring always lies on the surface of a hypersphere. Left: A high-dimensional ring over a hypersphere with 2 sinusoids and 6 von Mises functions with width parameter, $\kappa = 2$, plotted in a coordinate system made of the three first latent tuning functions ($c_1$, $c_2$ and $c_3$). Right: The same ring when the von Mises functions are infinitely broad ($\kappa = 0$). c) Example single-unit tuning functions from a network constructed using the latent tuning functions in a). Different colors represent different single units. d) The total ring length as a function of the embedding dimension, for different width parameters of the von Mises function. For broad tuning (low $\kappa$), increasing the embedding dimension shortens the ring. Note that this is due to our normalization scheme, which forces the ring to lie on a hypersphere (see right side of panel b). Conversely, increasing the embedding dimension of the ring when the tuning functions are relatively narrow will monotonically lengthen the ring. This is reminiscent of the theoretical result that increasing the density of narrow tuning functions tiling the stimulus space could increase Fisher information [33].
(PDF)

**S4 Fig. Low-D RNNs have a more robust basin of attraction.** We constructed 400-unit RNNs to embed either a 2-dimensional (top) or an 8-dimensional ring (bottom) of radius 12 and infinitely many fixed points (continuous attractor). We then simulated the RNNs from 100 random initializations. The initial conditions were drawn randomly from a spherical multivariate Gaussian distribution with a standard deviation of 12 in all dimensions. a) The initial state (blue) and end states (red) after 2 seconds of network activity plotted in a subspace spanned by the first three principal components for the two RNNs. b) The deviation (Eq 17) of neural states as a function of time during each simulation normalized by the radius. Initial deviations are very large, but the network state rapidly approaches the region of the state space near the ring. For the 2-dimensional ring (top), the end states coincide with the ring, suggesting that the ring acts as a global attractor. For the 8-dimensional ring, end states deviated more from the target manifold, suggesting that higher-dimensional rings are less robust.
(PDF)

**S5 Fig. Performance with respect to the number of setpoints.** We analyzed how the number of setpoints impacts the degree to which engineered RNNs can capture target manifolds and dynamics. We constructed the RNN to have O(8,6) with drift amplitude of 0.1 rad/s over a

ring of radius 12. a) Violin plot showing the mean squared error between the measured and target drift function as a function of number of setpoints. Inset: Target drift function (red) and the corresponding measured drift function with 6 setpoints (black). b) Violin plot showing the deviation (Eq 17) of neural states as a function of number of setpoints normalized by the radius. Results are shown for 20 simulations over a 5 second period, with deviation calculated from 20 evenly spaced initializations. Inset: The initial state (blue), intermediate states (gray), and end states (red) of a network constructed from 6 setpoints, plotted in a subspace spanned by the first three principal components.
(PDF)

**S6 Fig. Ring radius, unit saturation, and deviation.** The radius of the ring is equal to the magnitude of the projection vectors. We quantified how the radius changes the operating point of units with respect to their saturation point, and impacts the robustness of a 400-unit RNN engineered for O(8,6). a) Projections of the first latent vector in the neural state space (abscissa) passed through a hyperbolic tangent function for rings with different radii (ordinate). In rings with a large radius, more units operate near the saturation regime of their activation function. b) Violin plot showing the mean deviation (Eq 17) as a function of radius. For each radius value, 30 networks were simulated. Mean deviation was computed across 20 trajectories over 2 seconds. The line shows the means of the distributions of deviations for each radius.
(PDF)

## Acknowledgments

The authors wish to thank SueYeon Chung, Srdjan Ostojic, Ila Fiete, and Larry Abbott for helpful comments and discussions.

## Author Contributions

**Conceptualization:** Eli Pollock, Mehrdad Jazayeri.

**Data curation:** Eli Pollock.

**Formal analysis:** Eli Pollock.

**Funding acquisition:** Mehrdad Jazayeri.

**Investigation:** Eli Pollock.

**Methodology:** Eli Pollock.

**Project administration:** Eli Pollock, Mehrdad Jazayeri.

**Resources:** Mehrdad Jazayeri.

**Software:** Eli Pollock.

**Supervision:** Mehrdad Jazayeri.

**Validation:** Eli Pollock.

**Visualization:** Eli Pollock.

**Writing – original draft:** Eli Pollock, Mehrdad Jazayeri.

**Writing – review & editing:** Eli Pollock, Mehrdad Jazayeri.

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
