## [Decision Letter · Decision Letter 0]

11 Feb 2020

Dear Dr. Jazayeri,

Thank you very much for submitting your manuscript "Engineering recurrent neural networks from task-relevant manifolds and dynamics" for consideration at PLOS Computational Biology.

As with all papers reviewed by the journal, your manuscript was reviewed by members of the editorial board and by several independent reviewers. In light of the reviews (below this email), we would like to invite the resubmission of a significantly-revised version that takes into account the reviewers' comments.

We cannot make any decision about publication until we have seen the revised manuscript and your response to the reviewers' comments. Your revised manuscript is also likely to be sent to reviewers for further evaluation.

Sincerely,

Alireza Soltani

Associate Editor

PLOS Computational Biology

Daniele Marinazzo

Deputy Editor

PLOS Computational Biology

Reviewer's Responses to Questions

**Comments to the Authors:**

Reviewer #1: Review uploaded as an attachment

Reviewer #2: This is a very interesting manuscript that describes a new method for generating interpretable neural networks when the flow field of the dynamics can be specified. While the paper is well-motivated and shows convincing numerical results, I think that it requires a few clarifications.

***Main comments***:

-The success of the method relies on the hope that the nonlinear dynamics and the freedom in the weight matrix will succeed in reproducing the desired Jacobian through constraining it at a few select points that are high-dimensional expansions of points in the task manifold. The choice of these points could potentially improve or hurt the process, and yet it is not easy to give general guidance as to how to choose them. I think that the manuscript would be easier to read if this would be clarified early in the text, for example on lines 107-108, by stating that the Jacobian will only be constrained at a few points at which phi'(x) is evaluated. Besides a mild statement in the Method (lines 428-430), it looks like the issue of selecting the setpoints was not investigated much in the current draft. Is there an 'optimal' level of saturation of the units /scale of the projection vectors? Are there choices of the setpoints in the task manifold that make it harder to solve the dynamics --in particular, some setpoints seem to impose more structure in the weight matrix than others, like those situated on opposite branches of the star in Fig. 2b, which involve the same directions for the tangents U and the same eigenvalues, and yet different phi'(x). Are these specific points important to constrain, or does one gain extra freedom in the types of solutions for the weight matrix if one excludes these types of 'symmetric' points?

-Line 117, there is a mathematical expression: J_{RNN}=U \\Sigma U^T , that is only valid if J_{RNN} is normal with real eigenvalues/eigenvectors, while the following equation (3) does not make any assumption about the properties of the eigenvectors or eigenvalues. I think it would be less confusing to write the more general equation J_{RNN}=U \\Sigma U^(-1) on line 117. From what I understand, the J_{RNN} matrices that the authors constructed were not constrained to be normal (the statement made much later in the methods line 428 about the construction of the projection vectors from an orthogonal set does not necessarily imply that all eigenvectors of the final matrix are orthogonal, I think/verified). This point needs to be clarified. Also, the authors do not discuss how the choice of setting all eigenvalues to be real affects the solution -- purely decaying firing rates without oscillations can be somewhat unrealistic in the general case, and it would be desirable to discuss, or show, how the method could be extended to the case of complex eigenvalues.

-Line 500-523: here, doesn't the construction method used -- which implicitly assumes that the dynamics can independently get to steady-state in the input dimension, while the ring's dynamics occur in the other dimensions -- strictly require orthogonality between the input direction and the directions along which the dynamics is not decaying very fast? It would be good to introduce/clarify this earlier, rather than just mentioning later (line 524) that the input direction is chosen to be orthogonal to the ring, which sounds as if this was not compulsory.

-Is there a reason why the authors chose to match the Jacobian to the task manifold, rather than more directly matching the projection of dx/dt on some vector matrix U (hence constraining the flow field as the authors somewhat do for fixed points in eq. 11)? It seems that, in this case, we can also use the expression: U(dx/dt)= (1/tau) U(-x+W^{T} phi(x)+I) to write an equation of the type: A_{x0} W = B_{x0} +C_{x0} where A, B and C are constrained by the desired properties of the matrix dx/dt at the particular point x0. I can see that for linear dynamical systems, the Jacobian has an advantage over dx/dt: it is independent of x; but it is not formally the case for the nonlinear systems considered by the authors. Is the optimization less sensitive to the choices of the setpoints if matching the Jacobian rather than dx/dt? It would be good to discuss this.

-I think it would help the readers if Fig. 5 would show examples of reconstructing the tuning curves from the RNN's activities initialized at different points on the task’s manifold. Also, in the associated text, it would be good to clearly expose and differentiate the two kinds of embedding that are implemented: first, an embedding of the ring in n-dimensions, and second the embedding of this n-dimensional representation in the N dimensional space of the units' activities.

-It would be useful if the authors would display examples of weight matrices, and quickly investigate whether they seem similar to the weight matrices learned by backpropagation (e.g. similar to Cueva et al., ICLR 2019; by looking at whether units with similar preferred angle tend to wire positively together). This is a very easy step towards the goal, discussed by the authors, of comparing their networks to more classically trained RNNs.

-Could the authors clarify if the decoding weights used to recover theta (which in the current manuscript are evaluated numerically) could be analytically expressed as a function of the projection vectors? It seems to me that, at least at fixed points, this should be feasible through (i) getting a relation between phi(x) and x by setting the derivative of x to 0; (ii) expressing x in the basis of the projection vectors; and (iii) using the known embedding relation between the projection vectors and [cos(theta), sin(theta)].

-Line 216: clarify 'infinitely' many, you mean as many as the number of units?

-Line 342: I guess you only mean that the process you just discussed gives the minimal size of the network capable of creating dynamics over a particular ring *when using your method to tune the weights*.

***Miscellaneous corrections***:

The paper needs to be reviewed for typos. Several examples are identified below.

-eq. 6: subindex should be 1 and not 0, to agree with line 140 'for some number m points'

-fig 1 legend: 'MIddle left' -> 'Middle' (line 87) ; 'An RNN models' -> 'model' (line 91)

-line 117: should be a space after 'JRNN' before 'can'

-line 151: missing a 'to' in the sentence 'discrete fixed points can be used introduce error-correcting'

-fig 2a right: no input to the RNN, I think, in this figure.

-line 153: delete 'of' from 'When humans report of a previously…'

-line 182: missing 'similar' in 'adjustment to what' (-> ‘non-trivial adjustment similar to what we discussed previously’)

-line 185: grammar problem in the sentence 'a baseline can be added straightforwardly by an additional constraints to equation (6)…' for example, change 'an' to 'adding'

-line 186: typo in figure number 'Fig. 1c'? should be Fig. 2c

-line 263: add 'additional': 'the additional eigenvectors are the same as'

-line 279: 'red indicate' -> 'red colors indicate', or ‘red indicates’

-fig. 5b and lines 326-327: non-consistent mention of x and c as the coordinates of the sphere. Both of these symbols are already used to describe other variables in the text (the RNN dynamics for x and the tuning curves for c), so it would be desirable to use a different name for these coordinates (e.g. 'z')

-fig. 6a: what was the embedding dimension there?

-line 421: repeat that the 'setpoints' are the points where the jacobian will be specified

-line 440: add the word 'setpoints', for example, 'we first created a ring of setpoints by taking the cosine and sine of 64 evenly spaced values'

-line 450: extra = in the equation

-line 543: replace 'x' in the right-hand side by theta

-line 579: 'deviation from eq. 16', with 'deviation' typeset as math, reads better than 'deviation' typeset as text

-line 584: by 'number of the ring dimensionality', you mean 'the ring's spherical embedding dimensionality'? More generally, it would be good to make sure that throughout the text the spherical embedding vs. the embedding in the network's activities are labeled differently and clearly identified.

-The typesetting of the equations is not always consistent, cf. ’Jobj’ in line 109 and 120, or the 'deviation' measure which is at times written as text. Ultimately, fixing this can make the text nicer to read.

**Have all data underlying the figures and results presented in the manuscript been provided?**

Reviewer #1: Yes

Reviewer #2: Yes

PLOS authors have the option to publish the peer review history of their article (what does this mean?). If published, this will include your full peer review and any attached files.

Reviewer #1: Yes: Brian DePasquale

Reviewer #2: Yes: Laureline Logiaco
---

## [Decision Letter · Decision Letter 1]

6 Jun 2020

Dear Jazayeri,

Thank you very much for submitting your manuscript "Engineering recurrent neural networks from task-relevant manifolds and dynamics" for consideration at PLOS Computational Biology. As with all papers reviewed by the journal, your manuscript was reviewed by members of the editorial board and by several independent reviewers.

The reviewers appreciated the attention to an important topic. Based on the reviews, we are likely to accept this manuscript for publication, providing that you modify the manuscript according to the review recommendations. **More specifically, there are few minor issues raised by Reviewer # 2 that need to be addressed. To expedite the next round, we will not send out your revised manuscript for review and instead, make the final decision at the editorial level. **

Sincerely,

Alireza Soltani

Associate Editor

PLOS Computational Biology

Daniele Marinazzo

Deputy Editor

PLOS Computational Biology

[LINK]

Reviewer's Responses to Questions

**Comments to the Authors:**

Reviewer #1: All of my initial concerns have been addressed by the authors. In particular, I think that the rewrite of several later sections ("Rings embedded..." and "Limitations of RNN...") has greater improved the clarity of the work.

Reviewer #2: The authors have made a considerable amount of additional analyses and text clarifications which, in my opinion, now leads to an even greater article that is really nice to read.

My only minor remaining thought is that I was left partially unsatisfied with the analysis in Fig. 5e-f, though I want to leave it to the authors to decide whether they agree with my concerns and therefore want to improve this analysis in the final manuscript. Indeed, it is relatively expected that units with similar input weights would have similar tuning curves, as the input shapes the tuning. More interesting to me would be whether units with similar *output* weight vectors have similar tuning curves -- this would be expected in a 2d ring as the connectivity is likely symmetric, but for larger embedding dimensions it is unclear what happens. To answer this question, the analysis in Fig. 5e-f could be repeated using an 'output coupling' similarity matrix (J^T J) rather than the input coupling similarity matrix (J J^T) used by the authors. Alternatively, some examples of weight matrices could be presented with units ordered as for the sorted tuning similarity matrix.

In addition, the method used to sort the matrices in 5e should be indicated (for the analysis, it would make sense that some statistical technique such as hierarchical clustering would be used to sort one of them, and the second one would then also follow this order suggested by the data of the first one).

**Have all data underlying the figures and results presented in the manuscript been provided?**

Reviewer #1: Yes

Reviewer #2: Yes

PLOS authors have the option to publish the peer review history of their article (what does this mean?). If published, this will include your full peer review and any attached files.

Reviewer #1: Yes: Brian DePasquale

Reviewer #2: Yes: Laureline Logiaco
---

## [Editor Report · Decision Letter 2]

8 Jul 2020

Dear Dr. Jazayeri,

We are pleased to inform you that your manuscript 'Engineering recurrent neural networks from task-relevant manifolds and dynamics' has been provisionally accepted for publication in PLOS Computational Biology.

Best regards,

Alireza Soltani

Associate Editor

PLOS Computational Biology

Daniele Marinazzo

Deputy Editor

PLOS Computational Biology

---

## [Editor Report · Acceptance letter]

31 Jul 2020

PCOMPBIOL-D-19-02221R2 

Engineering recurrent neural networks from task-relevant manifolds and dynamics

Dear Dr Jazayeri,

I am pleased to inform you that your manuscript has been formally accepted for publication in PLOS Computational Biology. Your manuscript is now with our production department and you will be notified of the publication date in due course.

With kind regards,

Sarah Hammond
